# Estimating *Mycobacterium tuberculosis* transmission in a South African clinic: Spatiotemporal model based on person movements

Nicolas Banholzer[1], Keren Middelkoop[2,3], Juane Leukes[3], Ernest Weingartner [4], Remo Schmutz [1], Kathrin Zürcher[1], Matthias Egger [1,5,6], Robin Wood[3], Lukas Fenner [1]*

1 Institute of Social and Preventive Medicine, University of Bern, Bern, Switzerland, 2 Institute of Infectious Disease and Molecular Medicine, University of Cape Town, Cape Town, South Africa, 3 Desmond Tutu HIV Centre, Department of Medicine, University of Cape Town, Cape Town, South Africa, 4 Institute for Sensors and Electronics, University of Applied Sciences and Arts Northwestern Switzerland, Windisch, Switzerland, 5 Population Health Sciences, University of Bristol, Bristol, United Kingdom, 6 Centre for Infectious Disease Epidemiology and Research, University of Cape Town, Cape Town, South Africa

* lukas.fenner@unibe.ch

**Data availability statement:** Restrictions on the availability of personal data apply, but are

## Abstract

The risk of *Mycobacterium tuberculosis (Mtb)* transmission can be high in crowded clinics. We developed a spatiotemporal model of airborne *Mtb* transmission based on the Wells-Riley equation. We collected environmental, clinical and person-tracking data in a South African clinic during COVID-19, when community or surgical masks were compulsory and ventilation was increased. We matched person movements with clinical records to identify the spatiotemporal location of infectious TB patients. We modeled the concentration of infectious doses (quanta) and estimated the individual risk of infection. Over five days, video sensors tracked 1,438 clinic attendees. $CO_2$ levels were low (median 431 ppm, IQR 406 ppm–458 ppm); the quanta concentration was higher in the morning than in the afternoon, and highest in the waiting room. The estimated risk of infection per clinic attendee was 0.05% (80%-credible interval (CrI) 0.01%–0.06%). It increased with the number of close contacts with infectious patients and the time spent in the clinic, and was 1.3-fold (95%-CrI 1.2–1.4) higher in scenarios without mask use and 2.1-fold (95%-CrI 0.9–5.0) higher with pre-pandemic ventilation rates, emphasizing the importance of ventilation. Spatiotemporal modeling can identify high-risk areas and evaluate the impact of infection control measures in clinics.

## Author summary

In crowded indoor settings in high tuberculosis burden countries, there is a high risk of tuberculosis transmission. We developed a model to study how tuberculosis is spread

necessary to maintain the confidentiality of participants. The data is available upon reasonable request, contact: University of Bern, info.ispm@unibe.ch. All other data are included in the manuscript and accompanying supporting information. Code used for preprocessing, modelling and analysis, descriptive and simulation results are available on GitHub via https://github.com/nbanho/clinic-transmission.

**Funding:** The study was supported by a grant from the Swiss National Science Foundation (grant no. CRSK-3_190781). NB and LF are supported by the National Institute of Allergy and Infectious Diseases (NIAID) through grant no. 5U01-AI069924-05. ME is supported by special project funding from the Swiss National Science Foundation [grant no. 32FP30-189498]. The funders had no role in study design, data collection, data analysis, data interpretation, or writing of the report.

**Competing interests:** The authors have declared that no competing interests exist.

in a South African clinic during the COVID-19 pandemic when masks were mandatory and ventilation was improved. We tracked the movements of clinic attendees with video sensors and matched them with clinical records to find where and when infectious tuberculosis patients were present. We used this information to estimate the concentration of infectious particles in the air and the risk of infection for individuals. Over five days, the sensors tracked nearly 1,500 people in the clinic. Air quality was generally good, but the concentration of infectious particles was higher in the morning and peaked in the waiting room. The average risk of infection per person was found to be 0.05%. The risk increased with more close contact with infectious patients and longer time spent in the clinic. Without masks, the risk was 1.3 times higher. With lower ventilation rates, similar to pre-pandemic levels, the risk was 2.1 times higher. This modeling can help identify high-risk areas for tuberculosis transmission and evaluate infection control measures in healthcare facilities.

## Introduction

Tuberculosis (TB) remains one of the leading causes of death globally, with the South African region bearing a disproportionate share of the TB burden. After two years of COVID-19-related disruption of TB prevention and control measures, the negative impact of the COVID-19 pandemic is receding, as documented in the World Health Organization (WHO) 2023 global TB report [1]. However, progress was insufficient to meet global TB targets [1]. For example, investment in TB research was only about half of the USD 2 billion target [1].

*Mycobacterium tuberculosis* (*Mtb*), the causative agent of TB, is transmitted via respiratory particles in the exhaled air of infectious persons [2,3]. *Mtb* is primarily carried in small particles of $1-7\mu m$ [4], which can survive in the air for multiple hours [5]. Survival of airborne pathogens is greater in crowded, poorly ventilated indoor environments [2,6–9]. Healthcare facilities are high-risk settings where infectious individuals are more likely to visit [10]. A modeling study in South Africa estimated that 4% to 14% of adult TB cases originate from primary care clinics [11].

The Wells-Riley model [12] is widely used to estimate the risk of airborne transmission in a variety of indoor settings [13–17], including primary care clinics [18,19]. The model estimates the risk of infection based on the number of infectious individuals in space, the generation rate of infectious quanta (doses of pathogen-carrying particles), the breathing rate per person, and the outdoor air supply rate, which is often estimated from indoor $CO_2$ levels [13,20,21]. The Wells-Riley model and its variations [20] assume a well-mixed airspace, meaning that the quanta concentration is the same throughout the room. However, the concentration is typically higher near the infectious source [8,22,23], and previous studies have shown that the risk of TB infection is associated with proximity to infectious individuals [24,25].

We developed a spatiotemporal extension of the Wells-Riley model to model the concentration of infectious quanta in space and time. We combined clinical and person-tracking data to determine the spatiotemporal location of TB patients and susceptible attendees at a primary care clinic in South Africa for five days in October/November 2021. We measured indoor $CO_2$ levels to model the spatial diffusion and removal of infectious quanta over time and estimated the risk of infection per clinic attendee. Finally, we evaluated the impact of infection control measures during the study in response to the COVID-19 pandemic.

## Methods

### Ethics statement

The University of Cape Town Faculty of Health Sciences Human Research Ethics Committee (HREC/REF: 536/2020), the City of Cape Town (Project ID: 8139), South Africa, and the Ethics Committee of the Canton of Bern (KEK/REF: 2019-02131), Switzerland, approved the study.

### Study design and setting

Based on a pilot study from 2019 in the same clinic [18,26], we collected environmental data (indoor $CO_2$ levels), clinical data (TB status), and video sensor data (person movements) for five days during the COVID-19 pandemic in October/November 2021 (October 13, 15, 25, and November 4, 5) at a primary care clinic in Cape Town, South Africa. The clinic offers both TB and HIV services and other basic clinical services, Monday through Friday, from 8 am to 4 pm. It is located in a large settlement of formal and semi-formal housing where TB and HIV are highly prevalent [27,28]. We defined three areas within the clinic (Fig 1): the waiting room (10.55 m × 5.50 m × 3.00 m), corridor (12.45 m × 2.20 m × 2.50 m), and TB room (4.75 m × 3.50 m × 3.00 m). The TB room was mainly used for the treatment of diagnosed TB patients but also for the screening of suspected patients with respiratory symptoms who were ultimately not diagnosed with TB. The clinic was naturally ventilated by opening all doors and windows throughout the day.

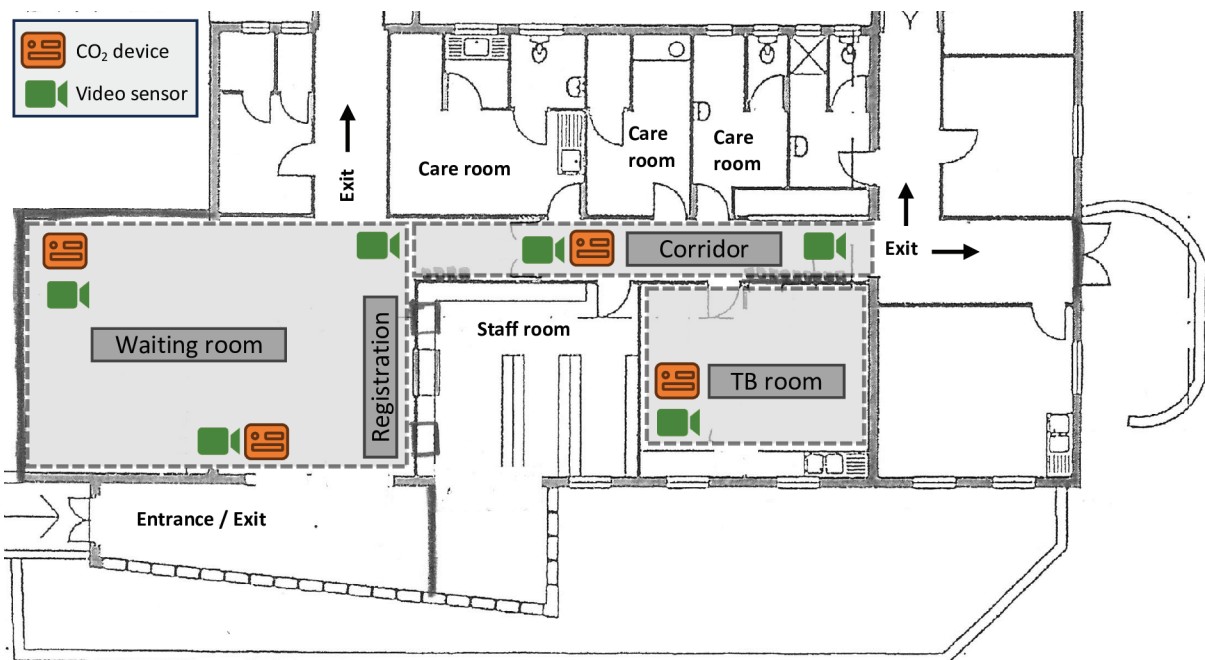

**Fig 1. Schematic view of the clinic**. Patients typically enter the clinic through the entrance on the bottom left, register at the reception, then wait in the waiting room or along the corridor until they see a doctor in one of the care rooms or the TB room. They exit the clinic through one of the exits in the waiting room or at the end of the corridor. Colored icons show the placement of the video sensors to track person movements and the devices monitoring $CO_2$ levels.

## Data collection

We collected environmental and clinical data and data on the movements of patients in the clinic. Four devices monitored indoor $CO_2$ levels ($CO_2$ Logger CL11, Rotronic, Bassersdorf, Switzerland) in the waiting room, corridor, and TB room (Fig 1), in parts per million (ppm) at 1-min intervals. On October 13, the corridor's $CO_2$ measurements were missing due to a malfunctioning device and the waiting room's $CO_2$ measurements were used for the corridor because their $CO_2$ measurements were similar on the other days (see time-varying $CO_2$ levels and number of persons per room by clinic day in Figs A–B in S1 Text, Sect A).

We extracted clinical data from the electronic patient registry for all patients who attended the clinic during the study. Clinical records included the date and time of arrival, TB diagnostic results, and date of TB treatment initiation. We defined infectious patients as individuals who tested bacteriologically positive for TB during the study or who had received TB treatment for less than four weeks. Patients on treatment for more than four weeks or with recently completed treatment were considered neither infectious nor susceptible. We also defined suspected TB patients as individuals who were tested for TB following symptom screening, but who were not diagnosed with TB.

We used an anonymous person tracking system using video sensors (Xovis, Zollikofen, Switzerland) to track the movements of people (clinic staff, patients, and other visitors) throughout the clinic at 1-second intervals (Fig 1). The resulting time-stamped tracking data consisted of a person's height, their position recorded as x-y coordinates, and a unique ID for each person's track while in the clinic. People in the clinic could contribute multiple tracks if they moved out of the range of a sensor or were not recognized as a person. To address this, we created an R Shiny application (Fig C in S1 Text) to rejoin interrupted tracks as described in Sect B in S1 Text. We also labeled the tracks to identify the movements of clinic staff. Fig 2 shows four examples of person movements after rejoining interrupted tracks.

## Spatiotemporal modeling approach

Our spatiotemporal model is based on the Wells-Riley equation [12], which estimates the risk (probability in %) of *Mtb* infection *P* as

$$P = \frac{C}{S} = 1 - \exp\left(-\frac{Ipqt}{Q}\right),\tag{1}$$

where *C* is the number of diseased cases, *S* is the number of susceptible cases, *I* is the number of infectious people in the indoor space, *p* is the breathing rate per person ($m^3\ h^{-1}$), *q* is the quantum (infectious dose) generation rate (quanta $h^{-1}$), *t* is the exposure time (h), and *Q* is the ventilation rate ($m^3\ h^{-1}$). The probability of infection is estimated with a Poisson relation, considering the stochastic behavior of airborne infection, where one quantum corresponds to a 67% risk of infection. The main assumption of the Wells-Riley model is a well-mixed airspace, which means that the quanta concentration $N = Iq/Q$ (quanta $m^{-3}$) is the same everywhere in the room.

We extended the Wells-Riley model, allowing spatial variation in the quanta concentration by combining patient-tracking data with clinical records. This way, the spatiotemporal location of infectious patients who generate quanta could be determined. A detailed description of our extension is provided in Sect C in S1 Text, together with an illustrative example using simulated data (Fig D in S1 Text). In the following, we describe the six main components of our modeling approach (Fig 3).

**Date: 2021-10-15  I  ID: 8644**

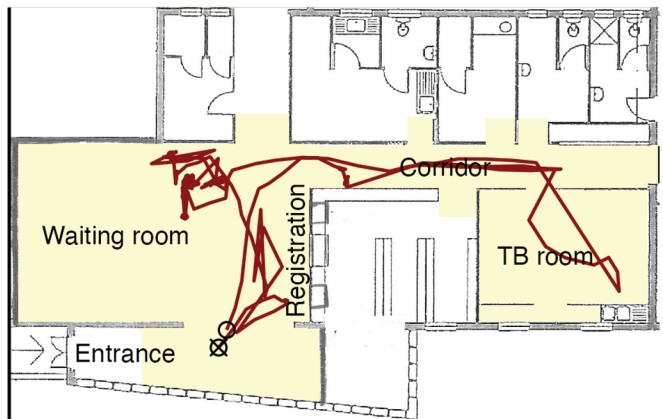

**Date: 2021-10-25  I  ID: 4649**

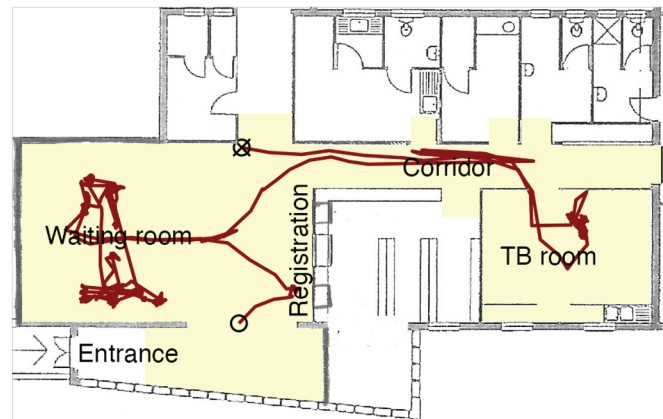

**Date: 2021-10-25  I  ID: 9526**

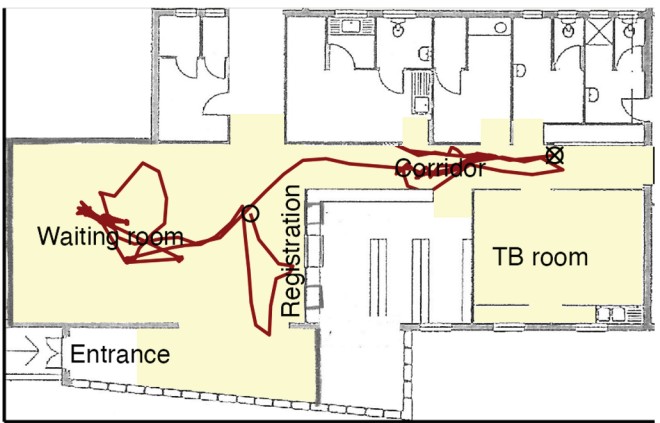

**Date: 2021-11-04  I  ID: 9841**

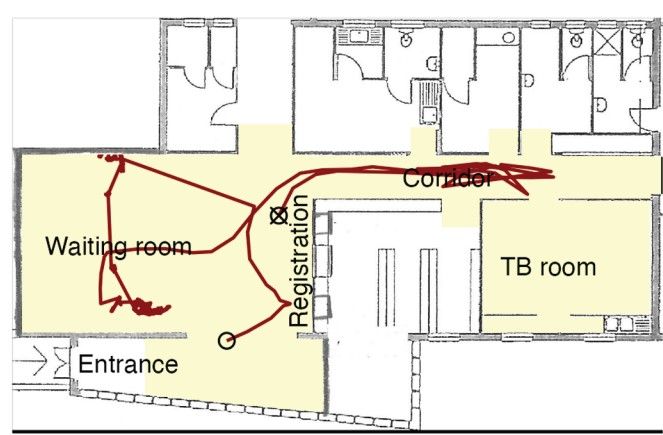

**Fig 2. Examples of person movements in the clinic.**

**Quanta generation.**  (1) We matched clinical records with person movements to identify the spatiotemporal location of infectious TB patients within the clinic. We recorded the timestamp when a person was detected in the registration area and matched this with the registration time in the clinical records. We excluded tracks with a total duration of less than five minutes in the clinic and a duration of less than five seconds in the registration area from the matching. We allowed for a maximum delay of 15 min between the track's timestamp in the registration area and the entry in the clinical database. This way, all diagnosed TB patients could be matched with person movements.

(2) In addition to diagnosed TB patients, we also assumed a number of undiagnosed (subclinical) TB patients. The number of undiagnosed TB patients may be proportional to the number of diagnosed TB patients [29–31]. Therefore, we modeled the number of undiagnosed TB patients with a multinomial distribution based on the counts of the number of diagnosed patients in the clinical data in October/November 2021 (Fig G in S1 Text): 6 days (12%)

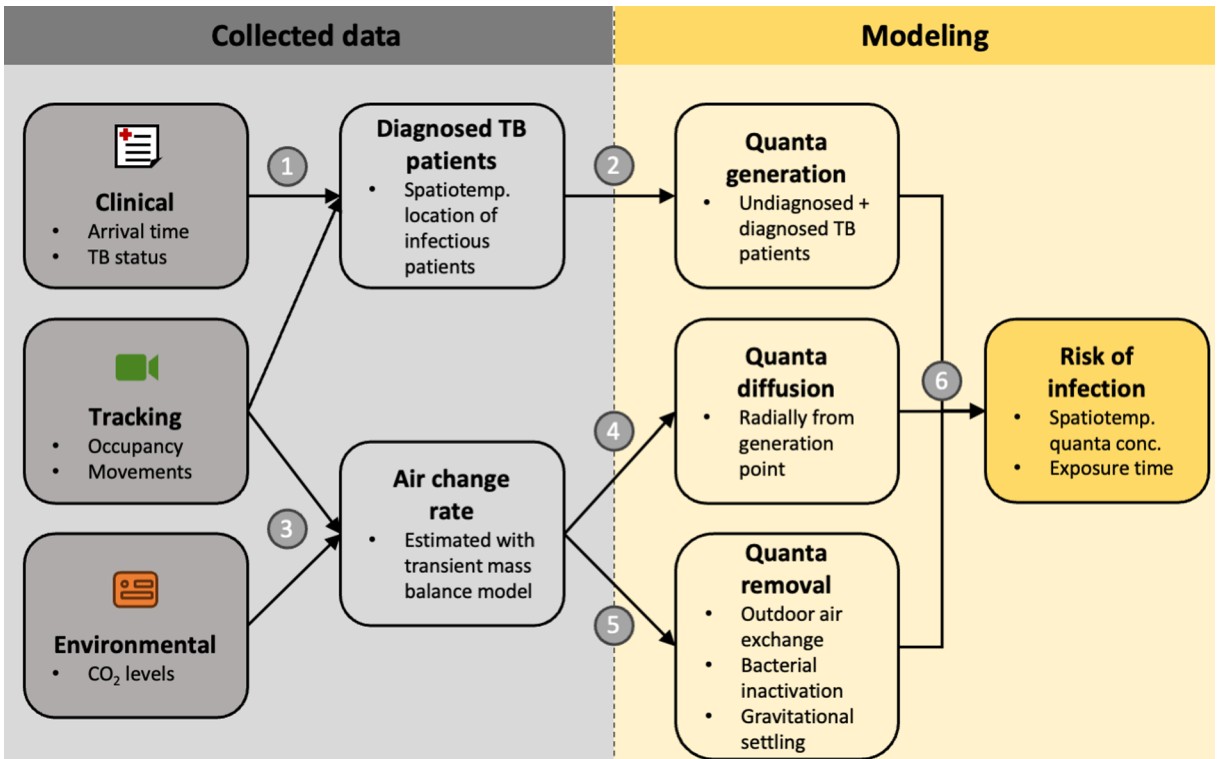

**Fig 3. Visual summary of the spatiotemporal modeling approach.** (1) Clinical records and person movements were linked based on their timestamps at the registration to determine the spatiotemporal location of diagnosed (infectious) TB patients. (2) Both diagnosed and randomly selected undiagnosed TB patients among clinic attendees generate infectious quanta. (3) Person-tracking and environmental data were combined to estimate the air change rate. (4) Air change rate was related to the speed at which quanta diffuses indoors. (5) The air change rate also determines quanta removal through outdoor air exchange, in addition to inactivation of *Mtb* in airborne particles... and removal through gravitational settling. (6) Quanta generation, diffusion, and removal determine the spatiotemporal quanta concentration, from which the personal risk of infection is calculated using the Wells-Riley equation.

with zero, 12 (41%) with one, 8 (28%) with two, 2 (7%) with three, and 1 (3%) with four diagnosed TB patients. Undiagnosed patients were randomly sampled among all clinic attendees with the same probability, excluding clinic staff who were not considered infectious. We further assumed that diagnosed and undiagnosed TB patients generated quantum at the same rate $q$, modeled using a lognormal prior distribution considering two different activity levels (Fig E in S1 Text): median of 1.08 quanta h$^{-1}$ (95%-credible interval [CrI] 0.003 quanta h$^{-1}$ – 386 quanta h$^{-1}$) while sitting and 2.81 quanta h$^{-1}$ (95%-CrI 0.008 quanta h$^{-1}$ – 987 quanta h$^{-1}$) while walking [32–34]. Since mask use (community or surgical masks) was mandatory for all clinic staff and attendees during the study, we assumed a mean reduction in $q$ by 75% (95%-CrI 56% – 86%) [19,35].

**Quanta diffusion.** (3) We calculated time-varying room occupancy from the video sensor data and combined it with the $CO_2$ measurements (Figs A–B in S1 Text) to the estimate air change rate during the morning (8 am to 12 am) and afternoon (12 am to 4 pm) of each clinic day using a transient mass balance model [36].

(4) The air change rate was related to the diffusion of quanta using, as an approximation, the empirical relationship between the eddy diffusion coefficient and the air change rate in $CO_2$ particles [37,38]. Since airflow was not monitored in our study, we assumed that the quanta diffused radially from where it had been generated.

**Quanta removal.** (5) Quanta removal occurs through the replacement of contaminated indoor air with outdoor air, inactivation of bacteria in airborne particles, and gravitational settling. The removal rate is thus the sum of the air change rate, the inactivation rate of *Mtb*, and the gravitational settling rate. We modeled the bacterial inactivation rate of *Mtb* using a lognormal prior distribution (Fig G in S1 Text) with a median of $1\,h^{-1}$ (95%-CrI $0.1\,h^{-1}$ – $7.1\,h^{-1}$) [5,39–41]. Particles with *Mtb* in the size range of $1-7\mu m$ [4] have a settling velocity of $3.5 \cdot 10^{-5} - 1.5 \cdot 10^{-3} m\,s^{-1}$ in still air [22], which we modeled using a Gamma distribution with mean $7.7 \cdot 10^{-4} m\,s^{-1}$ and standard deviation $3.7 \cdot 10^{-4} m\,s^{-1}$. Considering a drop height of $1.7\,m$ (average height of visitors), the median gravitational settling rate was assumed to be $1.5\,h^{-1}$ (95%-CrI $0.5\,h^{-1}$ – $3.5\,h^{-1}$).

**Risk of infection.** (6) The quanta concentration was computed separately for the waiting room, corridor, and TB room. Each room was discretized into a grid of cubic cells, each cell covering a squared area of $0.0625\,m^2$. We assumed that the quanta concentration was vertically well-mixed and computed the quanta concentration $N_{r,c,t}$ in room $r$ and cell $c$ at time $t$ as

$$\underbrace{N_{r,c,t}}_{\text{new conc.}} = \left( D\Delta \big( \underbrace{N_{r,c,t-1}}_{\text{old conc.}} + \underbrace{I_{r,c,t} \cdot q \cdot 1\,s \cdot V_{r,c}^{-1}}_{\text{generation}} \big) \right) \cdot \underbrace{\exp\left(-(AER_{r,t} + \lambda + k) \cdot 1\,s\right)}_{\text{removal}}. \qquad (2)$$

$$\underbrace{\phantom{\left( D\Delta ( N_{r,c,t-1} + I_{r,c,t} \cdot q \cdot 1 s \cdot V_{r,c}^{-1} ) \right)}}_{\text{diffusion}}$$

where $D_t$ is the diffusion constant ($m^2\,s^{-1}$) varying by daytime, $\Delta$ is the Laplace operator (second-order differential operator), $I_{r,c,t}$ is the number of infectious individuals, $V_{r,c}$ is the room-specific cell volume ($m^3$), $q$ is the quantum generation rate (quanta $s^{-1}$), $AER_{r,t}$ is the room-specific air change rate varying by daytime ($s^{-1}$), $\lambda$ is the bacterial inactivation rate ($s^{-1}$), and $k$ is the gravitational settling rate ($s^{-1}$). We modeled the spatiotemporal quanta concentration during clinic hours from 8 am to 4 pm and updated it every second, corresponding to the frequency of the person-tracking data.

The patient-specific risk of infection depends on the cumulative exposure to infectious quanta and was computed using the Wells-Riley equation as

$$P = 1 - \exp\left(-\sum_r \sum_c \sum_t N_{r,c,t} \cdot \mathbb{I}_{r,c,t} \cdot p_a \cdot 1\,s\right), \qquad (3)$$

where $\mathbb{I}_{r,c,t}$ indicates whether the person was in cell $c$ of room $r$ at time $t$, and $p_a$ ($m^3\,s^{-1}$) is the breathing rate for activity level $a$. We distinguished between walking ($p_{\text{walk}} = 1.33\,m^3\,h^{-1}$) and sitting activities ($p_{\text{sit}} = 0.51\,m^3\,h^{-1}$) [42], based on whether the the person had moved by more or less than $0.25\,m\,s^{-1}$. In line with a previous modeling study [19], we assumed that the mandatory use of community and surgical masks during our study had only effects on the exhalation of *Mtb* particles and no effect on *Mtb* infection of the person wearing the mask.

## Hypothetical scenarios

Data were collected during the COVID-19 pandemic when infection control measures were in place. To model their effectiveness, we considered two hypothetical scenarios: 1) no mask-wearing, i.e., assuming no reduction in the quantum generation rate; and 2) pre-pandemic ventilation conditions, i.e., assuming an air change rate of 6 air changes per hour, corresponding to the average rate in the pilot study in 2019 in the same clinic [18]. Furthermore, we examined the impact of different assumptions about the number of infectious patients in the

clinic: 1) we considered patients clinically suspected of having TB as infectious because these patients may generate aerosolized *Mtb* even when not diagnosed with TB at their first visit [31]; 2) we randomly selected infectious patients among all clinic attendees in proportion to the prevalence of TB in the South African population [30].

## Statistical analysis

We summarized clinical (diagnosed and suspected TB patients), environmental (time-varying $CO_2$ levels and air change rates), and person-tracking data (spatiotemporal number of tracks, visit time, and close-contact encounters) using descriptive statistics. We modeled the spatiotemporal quanta concentration with 5,000 Monte Carlo simulations, considering uncertainty in modeling parameters (Sect D in S1 Text). In each simulation, we computed the individual risk of infection based on the cumulative quanta exposure during the clinic visit. We computed the average quanta concentration by daytime (morning: 8 am to 12 am, afternoon: 12 am to 4 pm) and the mean risk of infection per clinic attendee across simulations. The scenario-specific infection risk distribution was shown with a histogram and summarized with the mean and 80%-credible interval (CrI). The mean risk of infection per clinic attendee was categorized as low (<0.1%), medium (0.1-1%), and high (>1%). We used a Bayesian Beta regression model to estimate the odds ratio (OR, posterior mean and 95%-CrI) of the modeled risk of TB infection in association with the visit time, duration of close contact with other clinic attendees, and frequency of close-contact encounters (proximity to another attendee <1 m for a duration of >1 min). Analyses were performed in R (version 4.3.1) and Stan (version 2.26.1). Additional results not reported in the manuscript are shown in Sect E in S1 Text.

## Results

During one week (five open clinic days), 894 patients were registered in the clinical database: four patients were bacteriologically diagnosed with TB and 16 patients were suspected of having TB after symptom screening. Overall, 1,563 person movements were detected based on the video sensor data; 125 likely from clinic staff (movements from and to staff rooms) and 1,438 from clinic attendees (registered patients and other visitors). Most movements were detected in the waiting room (Fig 4A) and during the morning (Fig 4B). Attendees spent about half an hour in the clinic (median 25 min, interquartile range (IQR) 13 min–46 min), mainly in the waiting room (Fig 4C). Time spent in the corridor and TB room varied considerably as many patients just walked through the corridor and the TB room was reserved primarily for diagnosed or suspected TB patients. Close-contact encounters were frequent (median 5, IQR 2–9 per attendee) and attendees were in close contact with at least one other person for more than half of the duration of their visit (median 68%, IQR 41%–87%). The $CO_2$ levels in the clinic were low (median 436 ppm, IQR 410 ppm–462 ppm) and the corresponding air change rates were high throughout the study (Fig 4D), with a median of 17 air changes per hour (IQR $9\,h^{-1}$–$23\,h^{-1}$) in the densely occupied waiting room. $CO_2$ levels varied slightly within and between clinic days, largely corresponding to changes in room occupancy during the day (Figs A–B in S1 Text), and possibly small variations in ventilation conditions between days. Median temperature in the clinic was 22°C (IQR 21°C–23°C) and relative humidity was 55% (IQR 47%–61%).

The quanta concentration was higher in the morning than in the afternoon, and higher in the waiting room than in the corridor or TB room of the clinic (Fig 5). An exposure of one hour to the mean quanta concentration in the middle of the waiting room would translate to a 0.20% risk of infection in the morning compared to a 0.04% risk in the afternoon. The mean

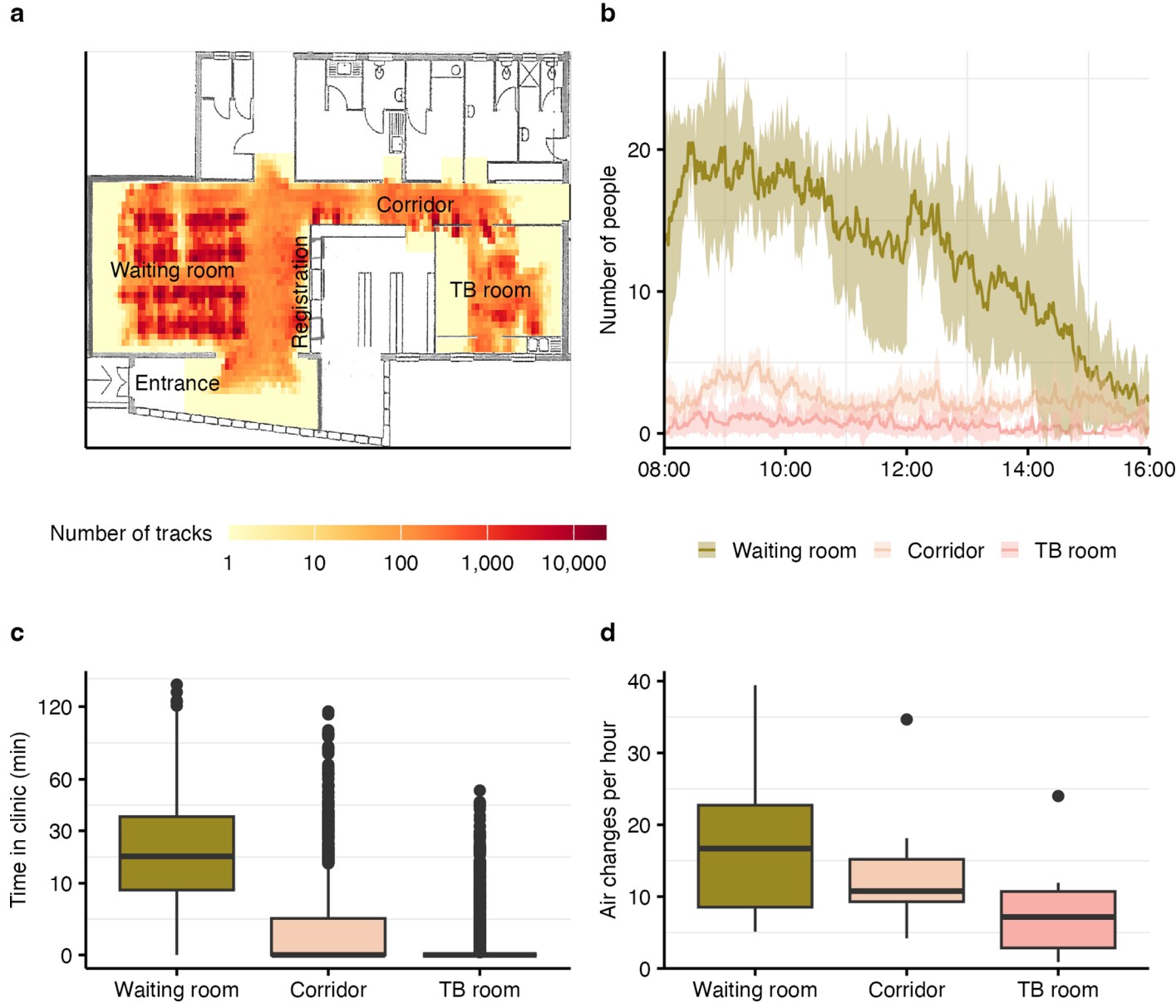

**Fig 4. Person tracks and environmental conditions by clinic area.** Across clinic days and for the clinic waiting room, corridor and TB room: **(a)** total number of tracks recorded per unit space, **(b)** number of people in the clinic over time, **(c)** time per clinic visit, and **(d)** air change rates. Lines and ribbons show the mean plus/minus standard deviation. Boxplots show the medians as lines, interquartile ranges as boxes, ranges as whiskers, and outliers as dots.

risk of infection per clinic visit was 0.05% (80%-credible interval (CrI) 0.01%–0.06%), corresponding to a risk of 0.1% per hour (80%-CrI 0.01%–0.21%). S1 Video shows an animated sequence of the spatiotemporally varying quanta concentration as an infectious patient enters the clinic on the morning of October 25 and moves through the waiting room. It can be seen that the quanta concentration was higher near the infectious patient, following their position and movement. The quanta diffused quickly because the diffusion constant was related to the air change rate, which was generally high because all doors and windows were typically open during clinic hours.

The modeled individual risk of infection was associated with the number of close contacts and the time spent in the clinic. A doubling of close-contact encounters increased the odds of infection by 17% (OR 1.17, 95%-CrI 1.04–1.30). Doubling the time spent in the clinic increased the odds of infection by 12% (OR 1.12, 95%-CrI 1.03–1.22). The time spent in close contact was not significantly associated with the individual risk of infection (OR 1.02, 95%-CrI, 0.94–1.11).

The hypothetical scenarios assuming no mask use and pre-pandemic ventilation rates are shown in Fig 6. If none of the clinic attendees or staff had worn masks, the mean risk of infection per visit would have been 0.07% (80%-CrI 0.01%–0.08%), an increase of 30% (OR 1.3, 95%-CrI 1.2–1.4). If the air change rate had been the same as before the COVID-19 pandemic, the mean risk of infection would have been 0.09% (80%-CrI 0.01%–0.10%), an increase of 110% (OR 2.1, 95%-CrI 0.9–5.0). Without mask use and with pre-pandemic ventilation rates, the mean risk of infection would have been 0.12% (80%-CrI 0.02%–0.12%), an increase of 170% (OR 2.7, 95%-CrI 1.1–6.6). The scenarios show an increase in the number of clinic attendees with a high risk of infection (Fig 6).

In sensitivity analyses, we varied assumptions about the number of infectious persons in the clinic. Assuming that the number of infectious TB patients reflected the prevalence of TB in the South African population, the mean risk of infection would have been slightly higher (0.06%, 80%-CrI 0.01%–0.10%). Further, assuming that both diagnosed TB patients and clinic attendees suspected of having TB were infectious, the mean risk of infection would have been twice as high (0.11%, 80%-CrI <0.01%–0.18%).

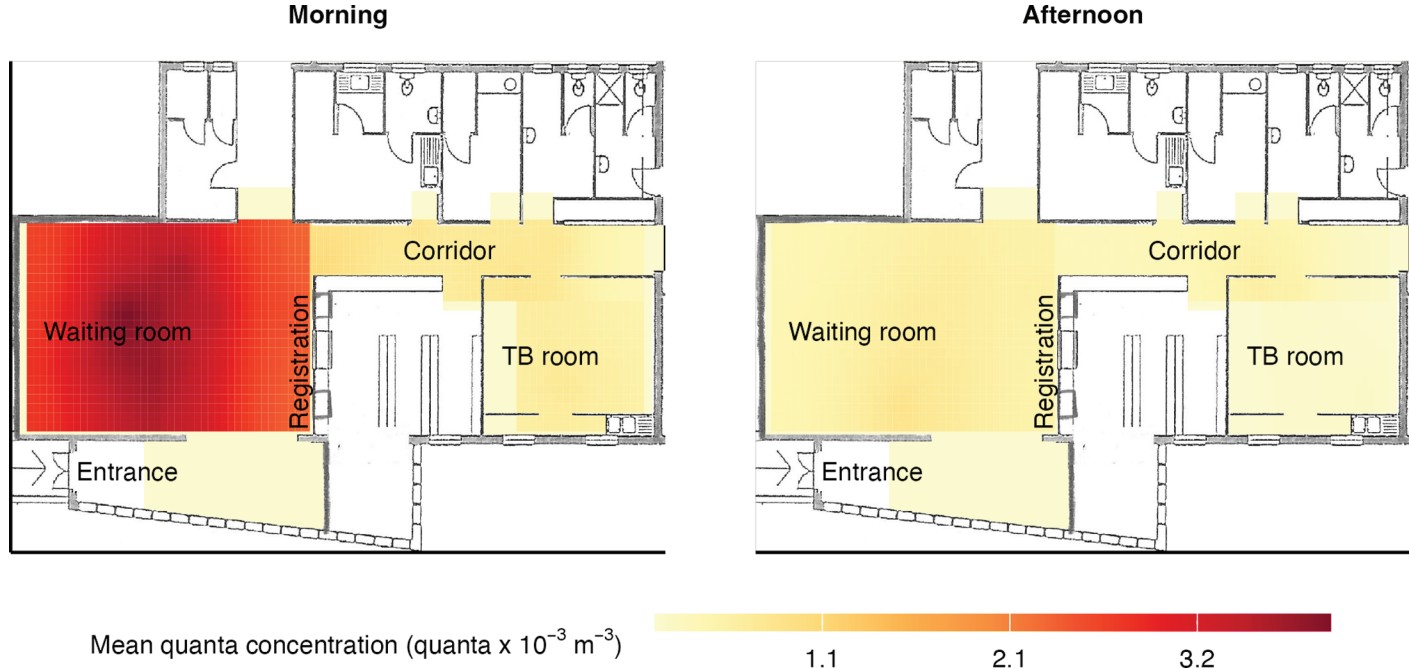

**Fig 5. Spatiotemporal quanta concentration by clinic area.** Model-estimated average quanta concentrations during clinic days in the morning and afternoon in the clinic's waiting room, corridor and TB room of the clinic.

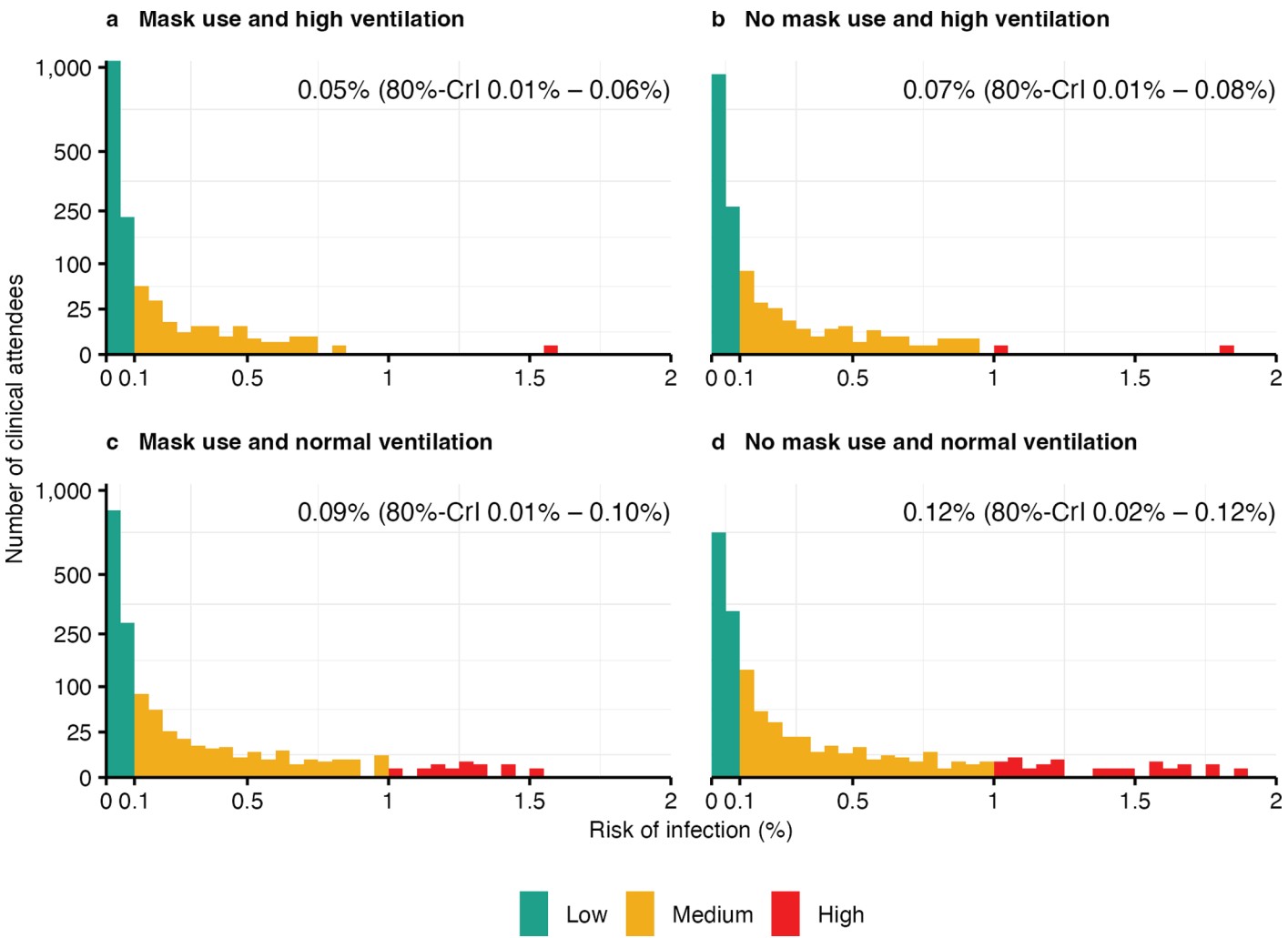

**Fig 6. Impact of infection control measures.** Mean risk of infection per clinic attendee (median visit time 25 min, IQR 13 min–46 min): **(a)** with mask use (community or surgical masks) and high ventilation as observed during the COVID-19 pandemic, **(b)** without mask use and high ventilation, **(c)** with mask use and normal ventilation as observed before the COVID-19 pandemic **(d)** without mask use and normal ventilation. Low risk: <0.1%, Medium: 0.1–1%, High: >1% risk of infection.

## Discussion

We expanded the Wells-Riley equation to a novel spatiotemporal model based on environmental, clinical, and patient movement data. We applied the model to real-world data from a South African primary care clinic collected in 2021 during the COVID-19 pandemic to examine the risk of *Mtb* infection among clinic attendees. We estimated an average risk of infection per clinic attendee of 0.05%. We found that clinic attendees who stayed longer and had more close contact encounters had a higher risk of infection. Without infection control measures (community/surgical masks and high natural ventilation), the risk of infection would have been almost three times higher. Our spatiotemporal model can be easily adapted to model airborne transmission risks of other respiratory infections such as SARS-CoV-2.

The COVID-19 pandemic has renewed interest in airborne transmission of respiratory infections in crowded indoor settings [8,9,43] and its modeling [33,44,45]. Many studies

[14–17] rely on the Wells-Riley transmission model [12], based on the pioneering work of William Wells, who proposed a role of droplets in airborne transmission in the 1930s [46], and Richard Riley, who developed the equation in the 1960s [12]. The Wells-Riley model assumes that the airspace is well-mixed, ignoring that closer proximity to infectious individuals most likely increases the risk of infection. Our spatiotemporal model is an extension of the Wells-Riley model, which allows the quanta concentration to vary both over time and space, and considers that the density of infectious particles is initially higher near the infectious source [8,22,23]. We found a significant association between the modeled risk of infection and the number of close-contact encounters, and the time spent in the clinic. More close contacts and extended visits make exposure to high doses of infectious particles more likely. Several previous studies have already indicated that prolonged close contact may be required to transmit respiratory infections [47–49].

South Africa has a high burden of TB and multi-drug-resistant TB [1]. Nevertheless, we estimated a lower risk of infection per clinic attendee in our primary care clinic under pandemic conditions with strict infection control measures in place. Assuming clinic visits amounting to 12 hours per year, we estimated a cumulative annual risk of infection of about 1.2%, which contrasts with an estimated risk of 9% to 29% from before the pandemic in the same clinic [18]. The difference can be mainly attributed to the infection control measures that were in place during the COVID-19 pandemic. We estimated that without mask use and with pre-pandemic ventilation conditions, the risk of infection would have been more than twice as high. Patient triage and fixed appointments probably also contributed to a lower risk of *Mtb* transmission during the COVID-19 pandemic, which was not considered in our hypothetical scenarios. Other modeling assumptions may also have contributed. The risk estimate from our previous study before the COVID-19 pandemic [18] was based on TB prevalence in the South African population, rather than the TB status documented in the clinical data. modeling the risk based on TB prevalence or assuming that both diagnosed and suspected TB patients were infectious also resulted in a higher modeled risk of transmission.

A previous study estimated that between 4% and 14% of TB transmission in adults in a high TB burden, high HIV prevalence community in South Africa occurred in primary care clinics [11]. Other studies suggest that mask-wearing could reduce the risk of *Mtb* transmission by more than 50% [19,35]. Our findings also show that the risk of infection is lower with mask use (community or surgical masks), although the risk reduction estimated in our study is less pronounced. The reason for this is that the effect of mask use is compounded by the effect of very high natural ventilation rates, which determine both the diffusion and the removal of infectious quanta in the indoor space. The air change rates in the clinic typically ranged from 10 to 20 air changes per hour, which resulted in a quick diffusion and removal of infectious quanta from the indoor air. To put this into perspective, we recently modeled the risk of *Mtb* transmission in schools by comparing ventilation conditions in South Africa (1.5 air changes per hour), Switzerland (0.7), and Tanzania (13.7) [34]. Tanzania had a much lower estimated risk of *Mtb* transmission in schools than the other countries, largely because of better natural ventilation, with an air change rate comparable to the one measured in the primary care clinic in our study. We emphasize that our results do *not* suggest that masks are ineffective – there is a large body of evidence for their effectiveness – but that their effect is outweighed by the excellent ventilation conditions in the clinic during the COVID-19 pandemic.

Airborne respiratory transmission depends on environmental and patient-specific factors. Our model considered ventilation conditions and patient movements. However, other factors such as variation in infectiousness, temperature and humidity, or physicochemical properties of the infectious particles can further influence the generation, diffusion, and removal of

infectious quanta (for a detailed discussion of these factors, see Sect F in S1 Text). Further-more, computational fluid dynamics (CFD) models have been developed to examine airflow as a factor influencing the spatial spread of airborne pathogens [22,50–54]. However, CFD models are computationally expensive and primarily used in experimental and static environments, making their application to real-world settings such as clinics difficult. Moreover, our modeling approach required computational efficiency because many Monte Carlo simulations are needed to reflect uncertainty in modeling parameters.

Our study has several limitations. First, tracking patient movements with video sensors was challenging. It required considerable data processing and many interrupted tracks could not be reconnected. Therefore, some clinic attendees may have had longer visits and the risk of infection may have been underestimated. Second, several important assumptions had to be made regarding the generation, diffusion, and removal of infectious quanta. For example, the waiting room and corridor were treated as separate rooms to facilitate modeling and because we observed differences in room occupancy and time-varying $CO_2$ levels, but the door between these areas was typically open, allowing aerosolized *Mtb* (quanta) to spread from one area to the other. We also assumed that infectious particles move radially from the source and approach a steady state with uniform contamination of the airspace, which may depend on airflow. However, airflow is very difficult to measure. Furthermore, uncertainty in modeling parameters determining quanta generation and removal was often large and could be reduced in future work by incorporating, for example, additional information on patient characteristics from clinical records that are known to be associated with infectiousness [55] or susceptibility [56]. Third, we could not validate our modeling results with empirical findings. Future research could test the practical implementation of our model by examining whether clinic attendees with a high risk of infection were also subsequently more likely to be diagnosed with TB.

In conclusion, our spatiotemporal transmission model incorporates clinical, environmental, and person movement data to estimate the risk of infection per person in the indoor space. It can be easily adapted to model airborne transmission risk of other respiratory infections such as SARS-CoV-2. We showed that the risk of *Mtb* transmission in a primary care clinic in South Africa was related to the number of close contact encounters and time spent in the clinic. The risk of infection was low under pandemic conditions with strict infection control measures in place. Our spatiotemporal model could be used in future work to evaluate the impact of infection control measures, specifically interventions targeting patient flows to reduce prolonged close contact. Finally, our study highlights the paramount importance of effective ventilation in primary care clinical settings.

## Supporting information

**S1 Text: Appendix.** Includes supplementary text, tables and figures.
(PDF)
**S1 Video.** Animated sequence of the spatiotemporally varying quanta concentration as an infectious patient enters the clinic and moves through the waiting room during one morning.
(GIF)

## Acknowledgments

We are grateful to the clinical manager and the staff members at the primary care clinic. We also would like to thank the City of Cape Town, South Africa, for the use of one of their clinic facilities and their support. Our research findings and recommendations do not represent an official view of the city of Cape Town.

## Author contributions

**Conceptualization:** Nicolas Banholzer, Keren Middelkoop, Kathrin Zürcher, Lukas Fenner.

**Data curation:** Nicolas Banholzer, Keren Middelkoop, Juane Leukes, Remo Schmutz, Lukas Fenner.

**Formal analysis:** Nicolas Banholzer, Remo Schmutz.

**Funding acquisition:** Matthias Egger, Lukas Fenner.

**Investigation:** Keren Middelkoop, Juane Leukes, Ernest Weingartner, Kathrin Zürcher, Matthias Egger, Robin Wood, Lukas Fenner.

**Methodology:** Nicolas Banholzer, Ernest Weingartner, Remo Schmutz, Lukas Fenner.

**Project administration:** Nicolas Banholzer, Keren Middelkoop, Juane Leukes, Kathrin Zürcher, Lukas Fenner.

**Resources:** Keren Middelkoop, Juane Leukes, Matthias Egger, Robin Wood, Lukas Fenner.

**Software:** Nicolas Banholzer.

**Supervision:** Lukas Fenner.

**Validation:** Nicolas Banholzer, Kathrin Zürcher.

**Visualization:** Nicolas Banholzer.

**Writing – original draft:** Nicolas Banholzer, Lukas Fenner.

**Writing – review & editing:** Nicolas Banholzer, Keren Middelkoop, Juane Leukes, Ernest Weingartner, Remo Schmutz, Kathrin Zürcher, Matthias Egger, Robin Wood, Lukas Fenner.

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
