## [Decision Letter · Decision Letter 0]

5 Nov 2024

PCOMPBIOL-D-24-01011Estimating Mycobacterium tuberculosis transmission in a South African clinic: Spatiotemporal model based on person movementsPLOS Computational Biology Dear Dr. Banholzer, Thank you for submitting your manuscript to PLOS Computational Biology. After careful consideration, we feel that it has merit but does not fully meet PLOS Computational Biology's publication criteria as it currently stands. Therefore, we invite you to submit a revised version of the manuscript that addresses the points raised during the review process. Please submit your revised manuscript within 60 days. If you will need more time than this to complete your revisions, please reply to this message or contact the journal office at ploscompbiol@plos.org. Please include the following items when submitting your revised manuscript: * A rebuttal letter that responds to each point raised by the editor and reviewer(s). You should upload this letter as a separate file labeled 'Response to Reviewers'. This file does not need to include responses to formatting updates and technical items listed in the 'Journal Requirements' section below.* A marked-up copy of your manuscript that highlights changes made to the original version. You should upload this as a separate file labeled 'Revised Manuscript with Track Changes'.* An unmarked version of your revised paper without tracked changes. You should upload this as a separate file labeled 'Manuscript'. If you would like to make changes to your financial disclosure, competing interests statement, or data availability statement, please make these updates within the submission form at the time of resubmission. Guidelines for resubmitting your figure files are available below the reviewer comments at the end of this letter. We look forward to receiving your revised manuscript. Kind regards, Katerina KaouriGuest EditorPLOS Computational Biology Denise KühnertSection EditorPLOS Computational Biology Feilim Mac GabhannEditor-in-ChiefPLOS Computational Biology Jason PapinEditor-in-ChiefPLOS Computational Biology  **Journal Requirements:** **Additional Editor Comments (if provided):**The paper is interesting and its data very useful for the field but please correct the mathematical framework, as highlighted by reviewer 2.**Reviewers' comments:** Reviewer's Responses to Questions

**Comments to the Authors:**

Reviewer #1: In this manuscript, Banholzer et al. develop methods to quantify the spatiotemporal changes of risk within a clinic accounting for movement of individuals in the clinic, uncertainty in the number of infectious individuals and changes of environmental exposure as a function of heterogeneous shedding, exchange of air through ventilation and the corresponding diffusion of pathogens in the environment. This manuscript is very well written, and the methods described have potential to be used to study a wide variety of pathogens in many settings.

Comments:

1. Consider including a reference to Figure F from the appendix at the sentence “Four devices monitored indoor CO2 levels (CO2 Logger CL11, Rotronic, Bassersdorf, Switzerland) in the waiting room, corridor, and TB room (Figure 1), in parts per million (ppm) at 1-min intervals” in the Data collection part of the methods. There are a number of additional connections that can be made between the main text and the appendix.

2. There is some justification for simulating quanta separately for waiting room, corridor and TB room. Can you describe in more detail how the different areas (waiting room, corridor and TB room) were ventilated? How was air flow connected between the three areas? And what could have been the implications of treating them separately even though there may be some dependence between them?

3. In the description of the spatiotemporal risk model, it sounds like the risk is calculated by ‘cell’ by second throughout the day, but then this sentence (Methods; Statistical analysis) makes it sound like the risk is averaged over half days. “The average quanta concentration was computed by daytime (morning: 8 am to 12 am, afternoon: 12 am to 4 pm) and the risk of infection was summarized with the mean and 80%-credible interval (CrI)” Can you clarify what the temporal resolution of the risk that you used was to generate your results?

4. Figure F in Appendix Section D presents figures that illustrate CO2 levels by clinic area over time for each study day. There is very little description of this data, yet differences in the measurements during and between days seems worth describing, including discussing possible reasons for the observed temporal changes over time and by study day.

5. Appendix Section E.2 Pathogen specific factors – Are there other surrogate pathogens for which survival by temperature and humidity are known that could inform the assumptions of the model with Mtb?

Reviewer #2: This paper presents a spatiotemporal model of airborne disease transmission, which extends the Wells-Riley equations by incorporating environmental, clinical and person movement data. The model is applied to simulate the infection risk of Mycobacterium tuberculosis in a South African clinic, where detailed environmental and movement data were gathered over five days during the COVID-19 pandemic. The authors use a Monte-Carlo simulation to consider the uncertainty in model parameters.

This paper is clearly written with nice graphics. The novelty of this work is highlighted by the combination of spatiotemporal models with real-world data, alongside the incorporation of uncertainty using a Monte Carlo approach. In addition, the authors have developed an open-source app to reconstruct the tracks of person movement, which will facilitate future studies in this field. I found this work a valuable contribution to the disease modelling and policymaking communities, and the methodology presented has the potential to be applied to other airborne diseases and a variety of indoor settings.

However, there are several issues in the mathematical model that require the authors’ attention. Addressing these issues may require updates to the simulation results and a revised discussion.

Major comments:

1. Methods – quanta removal (units): It is weird that the inactivation rate of Mtb has a unit of quanta/h. It should have the unit of 1/h or 1/s, because the inactivation quanta number is also proportional to the local quanta concentration. The same issue applies to the air exchange rate (AER_t in Eq.2). For the second part of Eq. 2, we need to get a number before taking the exponential (and thus to obtain a fraction of the previous quanta concentration due to decay factors). With the current unit, we will get exp(quanta), which doesn’t make sense.

Although the unit used for the inactivation rate of Mtb is wrong, the values seem appropriate for the unit of 1/h. I checked one reference (Gannon et al., 2007, Res J Vet Sci), and I got an inactivation rate of 0.46/h instead of 0.46 quanta/h, presented in Fig. D in S1 Appendix. In this case, it is likely (and hopefully) that this unit issue won’t affect the results and the conclusion of this paper. The authors should verify their derivation of the inactivation rate and the air exchange rate, and correct the units in both the manuscript and appendix where necessary.

2. Methods – quanta removal (gravitational settling): The authors set the gravitational settling rate to 0, citing the particles with Mtb have a settling velocity of 3.5e-5 to 1.5e-3 m/s. This sentence doesn’t make sense. For instance, if we take the maximum value 1.5e-3 m/s, the particles generated by an infectious person standing at 1.7m only take ~20 minutes to fall on the ground, which seems to dominate over other factors. Even if we assume that the settling velocity is at the order of 1e-4 m/s, it still yields ~0.2/h, which is comparable to the inactivation rate mentioned above. I would suggest the authors either include a gravitational settling rate, or acknowledge this limitation and provide a short discussion on how gravitational settling would affect the results.

3. Methods – Risk of infection (Eq. 2): Below Eq. 2, the authors define I_t as the number of infectious individuals in space. It seems the space here indicates a cell space c, instead of the space of the entire domain. In this case, I would suggest the authors to first introduce they discretised the space into cells, and then state explicitly that the Eq. 2 is for each cell ‘c’. In addition, the authors need to add subscript ‘c’ for N_t, N_(t-1), I_t and AER_t.

Furthermore, the units in the Laplacian don’t match, N_(t-1) is a concentration with quanta/m^3 while the second term gives quanta. I believe the q in the second term should be divided by the volume of a cubic cell. However, this depends on the mixing assumption made by the authors (see comment 5). The authors should check their model and code, and revise Eq. 2 accordingly.

4. Methods – Risk of infection (mask efficiency): The authors didn’t consider the effects of masks on the inhalation of infectious aerosols in Eq. 3. While the authors have assumed a reduction in quanta generation by 75%, a similar reduction in inhalation should be applied. In the Discussion, the authors suggest that the masks are less effective in their case study, due to high ventilation rates. However, I believe this is because the authors didn’t consider the impact of masks on inhalation.

5. Methods – Monte Carlo simulation (2D or 3D): In the appendix, the authors state that the aerosols diffuse in (x,y) direction. It is unclear how the authors calculate the quanta concentration along the room height. Did you assume the quanta are well-mixed vertically? The authors should clarify this in the manuscript. In addition, maybe use ‘…cubic cells, with each cell covering an area…’, since a cubic cell has a volume instead of an area.

Minor comments:

6. Introduction – 2nd paragraph: A minor suggestion: rephrase ‘because infectious individuals are more likely to visit clinics’ to ‘where infectious individuals are more likely to visit’.

7. Methods – quanta generation – 1st paragraph: The authors allowed a maximum delay of 15 minutes between an individual was detected in the registration area and their entry in the database. This seems quite long for a small clinic. I was wondering why the authors set the threshold as 15 minutes?

8. Results – 2nd paragraph & S2 video: It would be helpful to add the dimensions along the two axes, and represent the location of the infectious individual using a distinct colour in the video.

9. Results – Fig. 4: In Fig. 4b, the vertical axis should be number of people instead of number of tracks? In Fig. 4c, the outliers for TB room seem a bit too many? Some explanation for this would be helpful.

10. Appendix: I appreciate the authors’ efforts in preparing a detailed appendix, which includes plenty of useful information. However, for a mathematical biology journal like PLOS Computational Biology, it might be useful to include key modelling assumptions and numerical details in the main text (instead of the appendix), which would improve readability and accessibility.

**Have the authors made all data and (if applicable) computational code underlying the findings in their manuscript fully available?**

Reviewer #1: Yes

Reviewer #2: Yes

PLOS authors have the option to publish the peer review history of their article (what does this mean?). If published, this will include your full peer review and any attached files.

Reviewer #1: No

Reviewer #2: No

 **Figure resubmission:**While revising your submission, please upload your figure files to the Preflight Analysis and Conversion Engine (PACE) digital diagnostic tool, https://pacev2.apexcovantage.com/. PACE helps ensure that figures meet PLOS requirements. To use PACE, you must first register as a user. Registration is free. Then, login and navigate to the UPLOAD tab, where you will find detailed instructions on how to use the tool. If you encounter any issues or have any questions when using PACE, please email PLOS at figures@plos.org. Please note that Supporting Information files do not need this step. If there are other versions of figure files still present in your submission file inventory at resubmission, please replace them with the PACE-processed versions. 
---

## [Decision Letter · Decision Letter 1]

24 Jan 2025

Dear Dr Fenner,

We are pleased to inform you that your manuscript 'Estimating Mycobacterium tuberculosis transmission in a South African clinic: Spatiotemporal model based on person movements' has been provisionally accepted for publication in PLOS Computational Biology.

Best regards,

Katerina Kaouri

Guest Editor

PLOS Computational Biology

Denise Kühnert

Section Editor

PLOS Computational Biology

Dear authors,

Both reviewers are happy with the revisions you have submitted so your paper is accepted. It is a nice and interesting piece of work. Congratulations!

Best wishes,

Katerina Kaouri.

Reviewer's Responses to Questions

**Comments to the Authors:**

Reviewer #1: Thank you for the update, nice work.

Reviewer #2: The authors have addressed all my previous comments in their revised manuscript. I would like to thank the authors for their efforts and congratulations on this nice piece of work.

**Have the authors made all data and (if applicable) computational code underlying the findings in their manuscript fully available?**

Reviewer #1: Yes

Reviewer #2: None

PLOS authors have the option to publish the peer review history of their article (what does this mean?). If published, this will include your full peer review and any attached files.

Reviewer #1: No

Reviewer #2: No

---

## [Editor Report · Acceptance letter]

PCOMPBIOL-D-24-01011R1

Estimating Mycobacterium tuberculosis transmission in a South African clinic: Spatiotemporal model based on person movements

Dear Dr Fenner,

I am pleased to inform you that your manuscript has been formally accepted for publication in PLOS Computational Biology. Your manuscript is now with our production department and you will be notified of the publication date in due course.

With kind regards,

Zsofia Freund
